# Perception is associated with the brain's metabolic response to sensory stimulation

**Mauro DiNuzzo[1†], Silvia Mangia[2†], Marta Moraschi[3], Daniele Mascali[1,4], Gisela E Hagberg[5], Federico Giove[1,6]\***

[1]Museo Storico della Fisica e Centro Studi e Ricerche Enrico Fermi, Rome, Italy; [2]Center for Magnetic Resonance Research, Department of Radiology, University of Minnesota, Minneapolis, United States; [3]Department of Radiation Oncology, Campus Bio-Medico University of Rome, Rome, Italy; [4]Dipartimento di Neuroscienze, Imaging e Scienze Cliniche, Università Gabriele D'Annunzio, Chieti, Italy; [5]Biomedical Magnetic Resonance, University Hospital Tübingen and High–Field Magnetic Resonance, Max Planck Institute for Biological Cybernetics, Tübingen, Germany; [6]Fondazione Santa Lucia IRCCS, Rome, Italy

**Abstract** Processing of incoming sensory stimulation triggers an increase of cerebral perfusion and blood oxygenation (neurovascular response) as well as an alteration of the metabolic neurochemical profile (neurometabolic response). Here, we show in human primary visual cortex (V1) that perceived and unperceived isoluminant chromatic flickering stimuli designed to have similar neurovascular responses as measured by blood oxygenation level-dependent functional magnetic resonance imaging (BOLD-fMRI) have markedly different neurometabolic responses as measured by proton functional magnetic resonance spectroscopy (1H-fMRS). In particular, a significant regional buildup of lactate, an index of aerobic glycolysis, and glutamate, an index of malate–aspartate shuttle, occurred in V1 only when the flickering was perceived, without any relation with other behavioral or physiological variables. Whereas the BOLD-fMRI signal in V1, a proxy for input to V1, was insensitive to flickering perception by design, the BOLD-fMRI signal in secondary visual areas was larger during perceived than unperceived flickering, indicating increased output from V1. These results demonstrate that the upregulation of energy metabolism induced by visual stimulation depends on the type of information processing taking place in V1, and that 1H-fMRS provides unique information about local input/output balance that is not measured by BOLD-fMRI.

**\*For correspondence:**
federico.giove@uniroma1.it

[†]These authors contributed equally to this work

**Competing interest:** The authors declare that no competing interests exist.

## Editor's evaluation

The authors demonstrate an intriguing dissociation of neurovascular (as measured with BOLD-fMRI) and neurometabolic (measured with fMRS) responses during perception. This is a thought-provoking study that makes one wonder about the neurophysiological origin of the signals we measure with human neuroimaging, especially fMRI. It will therefore be of great interest to the broad community of neuroimagers, as well as perception researchers.

## Introduction

In the brain, sensory stimulation is associated with a substantial increase of regional functional hyperemia (i.e., cerebral blood flow, CBF) as well as energy metabolism of glucose, the main cerebral energy substrate (*Siesjo, 1978*). The metabolic response to stimulation includes an oxidative component, as

measured by the cerebral metabolic rate of oxygen ($CMRO_2$), and a nonoxidative component, as reflected by lactate accumulation (*Mangia et al., 2009*). Cortical lactate levels have been shown to increase during visual stimulation, simultaneously to the acceleration of the malate–aspartate shuttle (MAS), a process termed aerobic glycolysis (i.e., lactate production independent of oxygen availability) (*Bednařík et al., 2015*; *Bednařík et al., 2018*; *Boillat et al., 2020*; *Fernandes et al., 2020*; *Lin et al., 2010*; *Lin et al., 2012*; *Mangia et al., 2007a*; *Schaller et al., 2013*; *Schaller et al., 2014*). Despite intense research, the neurophysiological mechanisms underlying the upregulation of glycolytic metabolism of glucose are still largely unknown (*Dienel, 2019a*). Most importantly, the impact of information processing to the metabolic response of the cerebral cortex to sensory stimulation has not been fully investigated thus far. In particular, nothing is known about the modulatory effect exerted by the perception of different stimuli on regional brain energy metabolism.

Sensory perception is thought to rely on the complex interplay of neural circuits that process information in a cortical layer- and area-mediated manner involving thalamocortical, intracortical, cortico-cortical, and corticothalamic feedforward/feedback loops (*D'Souza and Burkhalter, 2017*). Sensory stimuli transduced by sensory organs reach specific thalamic nuclei that relay information to primary sensory cortices, which in turn filter and eventually transmit information to secondary sensory areas (*Crick and Koch, 1995*). These transactions are dependent on the particular features of different incoming stimuli, thus it is possible that the relevant neurovascular and neurometabolic counterparts are correspondingly distinct (*Lauritzen, 2001*).

The thalamic lateral geniculate nucleus (LGN) mediates visual stimuli with temporal frequencies at least up to 90 Hz to the layer IV of V1 (*Gur and Snodderly, 1997*; *Herrmann, 2001*; *Norcia et al., 2015*; *Pastor et al., 2003*; *Regan, 1989*; *Vialatte et al., 2010*), which in turn relays to output layers II/III and V where temporal filtering occurs (*Sachidhanandam et al., 2013*), consistent with the notion that visual perception requires the activation of visual areas downstream V1 (i.e., secondary visual areas). In agreement with these arguments, it has been repeatedly reported that invisible visual flickering is still able to activate V1 even without any perceptual effects (*Alais et al., 2016*), as revealed by in vivo electrophysiology in nonhuman primates (*Gur and Snodderly, 1997*) as well as behavioral evidence (*Lorenceau, 1987*) and blood oxygenation level-dependent (BOLD) functional magnetic resonance imaging (fMRI) (*Jiang et al., 2007*) in humans. High (30 Hz) frequency visual stimulation has been found to selectively suppress multiunit activity (MUA) in cat V1 as compared to low frequency (4 Hz) visual stimulation (*Viswanathan and Freeman, 2007*). Importantly, local field potentials and tissue oxygen response, which directly contribute to the generation of the BOLD signals (*Logothetis et al., 2001*), were preserved at both frequencies.

In the present study, we combined BOLD-fMRI and proton functional magnetic resonance spectroscopy (1H-fMRS) in humans and exploited the well-known effect of temporal frequency on visual perception. Specifically, we examined the functional and metabolic responses of the primary visual cortex (V1) to perceived or unperceived isoluminant chromatic flickering stimulation obtained by using temporal frequency either below (7.5 Hz; PF, perceived flickering) or above (30 Hz; UF, unperceived flickering) the critical flicker fusion (CFF) threshold of ~15 Hz for rod-mediated vision (*Hecht and Shlaer, 1936*). Based on experimental evidence and metabolic modeling, we have previously proposed that the local input–output balance between neuronal synaptic/spiking (or subthreshold/suprathreshold) activity is a primary determinant in the upregulation of aerobic glycolysis (*DiNuzzo and Giove, 2012*; *DiNuzzo et al., 2011*; *DiNuzzo and Nedergaard, 2017*). We thus hypothesized that the loss of visual perception is accompanied by fundamental changes in the metabolic responses of human V1.

## Results

### Subjects perception of the visual stimuli

To achieve perceptual isoluminance between green and red color (necessary for the UF condition), we adjusted the brightness of the green color for each individual subject, which was remarkably similar across subjects (green/red brightness ratio 71.9% ± 1.2%, range 70.1–73.5%) (*Figure 1—source data 1*). After this procedure, 100% of the subjects confirmed that their perception of the 30 Hz frequency stimulus steadiness was equivalent to the resting condition. Overall, the subject's perception was a gray/colored checkerboard that in the colored squares showed either a fast green and red alternation

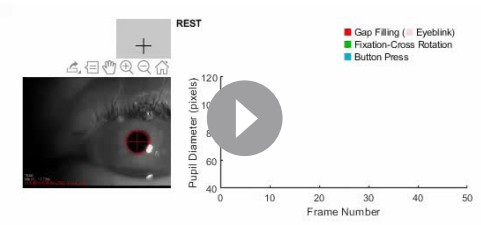

**Video 1.** The movie shows the physiological fluctuations of the pupil diameter, as well as the relevant behavioral responses, of a representative subject during a functional magnetic resonance imaging (fMRI) stimulation cycle, which included one unperceived flickering (UF; 30 s) and one perceived flickering (PF; 30 s) epoch.

https://elifesciences.org/articles/71016/figures#video1

during PF epochs, or a static yellow during UF epochs (*Video 1*). As a further confirmation, while in the scanner the subjects were unable to distinguish the 30 Hz red-green flickering checkerboard (used in the actual experiments) from a color-matched static yellow checkerboard (used for testing only). Specifically, the perception of the steady yellow color versus the 30 Hz red-green flickering was indistinguishable, as assessed by asking the subjects to guess the origin of the stimulus for 10 consecutive trials (average of correct responses 52% ± 16%, not different from chance level, p = 0.62). All subjects reported to distinctly perceive the green and red color when the checkerboard was flickering at 7.5 Hz. None of the subjects perceived the intrinsic flickering of the screen due to the refresh rate (60 Hz).

## Subjects attention and task performance

To examine the possible modulation of the BOLD signal by attention (*Watanabe et al., 2011*), we measured pupillometry and task-performance data during all experiments. All subjects maintained gaze on the fixation cross within 0.2–0.3° during all fMRI (*Figure 1—figure supplement 1*) and 1H-fMRS (*Figure 3—figure supplement 1*) epochs, with no difference in average gaze location between the different stimulations (one-way analysis of variance [ANOVA], p = 0.96 and p = 0.38, for fMRI runs 1 and 2, respectively, and p = 0.29 and p = 0.19 for 1H-fMRS runs 1 and 2, respectively). Pupil diameter, an index of the noradrenergic tone (*DiNuzzo et al., 2019*), was fairly stable at around 6.5–7 mm on average during all fMRI (*Figure 1—figure supplement 2*) and 1H-fMRS (*Figure 3—figure supplement 2*) epochs (one-way ANOVA, p = 0.99 and p = 0.98, for fMRI runs 1 and 2, respectively, and p = 0.98 and p = 0.96 for 1H-fMRS runs 1 and 2, respectively) across conditions (see also *Video 1*), indicating that the modulation of perception by noradrenaline (*Gelbard-Sagiv et al., 2018*) was minimal in our experimental conditions. The hit/miss ratio for the cross rotation task during the stimulation protocol was essentially 1.0, as expected due to the simplicity of the task, for the entire duration of the experiment. In particular, the delay of the response was not statistically different (one-way ANOVA, p = 0.09 and p = 0.77 for fMRI runs 1 and 2, respectively, and p = 0.35 and p = 0.51 for 1H-fMRS runs 1 and 2, respectively) for rest, PF and UF epochs (ranging, on average, between 360 and 460 ms), both during fMRI (*Figure 1—figure supplement 3*) and 1H-fMRS (*Figure 3—figure supplement 3*), confirming high and similar levels of subject's attention across conditions. There was not significant correlation between mean pupil diameter (two-tailed *t*-test, $q_{FDR}$ = 1 and $q_{FDR}$ >0.16 for fMRI runs 1 and 2, respectively, and $q_{FDR}$ >0.14 and $q_{FDR}$ >0.88 for 1H-fMRS runs 1 and 2, respectively) or gaze displacement (two-tailed *t*-test, $q_{FDR}$ = 1 and $q_{FDR}$ >0.59 for fMRI runs 1 and 2, respectively, and $q_{FDR}$ >0.13 and $q_{FDR}$ = 1 for 1H-fMRS runs 1 and 2, respectively) and task performance during both fMRI (*Figure 1—figure supplement 4*) and 1H-fMRS (*Figure 3—figure supplement 4*). Finally, in-scanner head motion during fMRI scans was minimal and not significantly different for all subjects across epochs (mean framewise displacement 0.25 ± 0.12 mm for rest, 0.24 ± 0.14 mm for PF, 0.23 ± 0.08 mm for UF; one-way ANOVA, p = 0.77). Overall, behavioral and physiological variables associated with attentional load were maintained at considerably constant levels in all subjects.

## Similar BOLD responses in V1 to PF and UF

To achieve the same BOLD response in V1 during PF and UF, we reduced the stimulation contrast for the 7.5 Hz condition to 75% relative to the 30 Hz condition (*Figure 1—figure supplement 5*). As expected, we found that the average BOLD time course (one-way ANOVA, p = 0.42), onset time (one-way ANOVA, p = 0.93) and time-to-peak (paired sample *t*-test, p = 0.29) did not differ between PF and UF (*Figure 1A, C and D*). Similarly, the BOLD change in the subject-matched spectroscopic volume-of-interest (VOI; on average consisting of 47% ± 9% of BA17, 21% ± 12% of BA18, and 16% ±

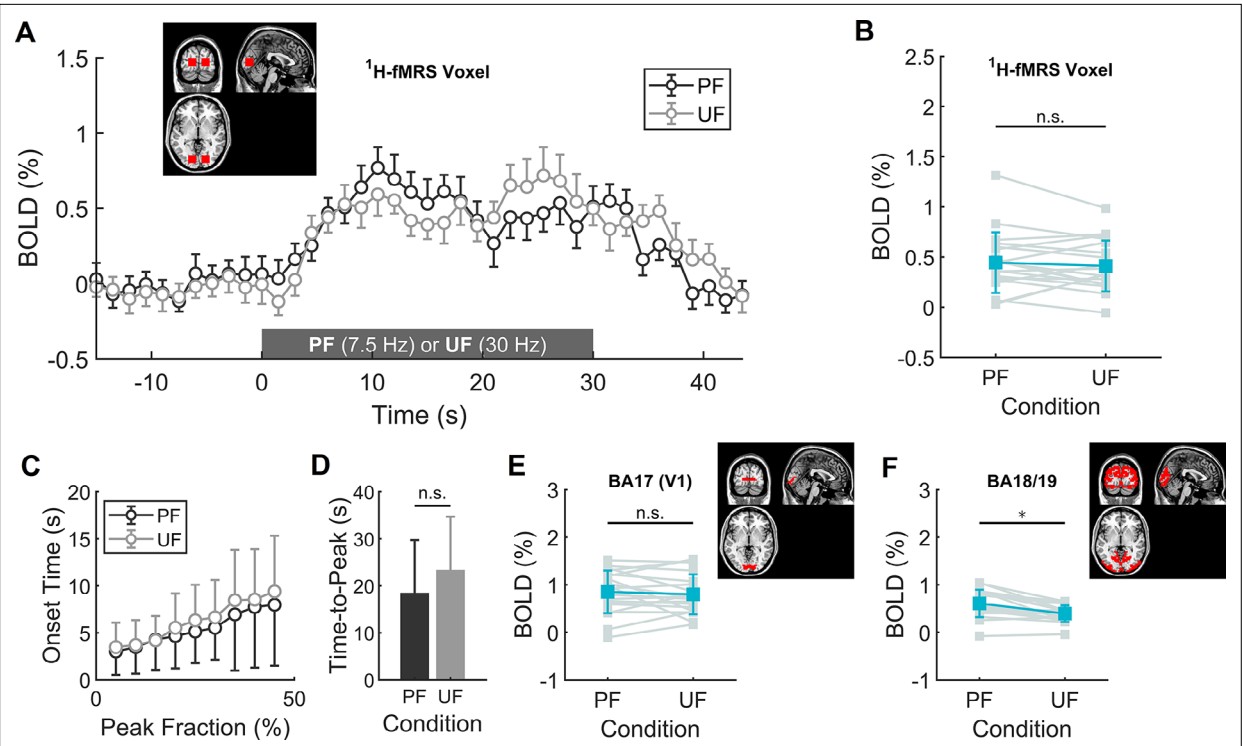

**Figure 1.** Main and differential effects of stimulation assessed by volume-of-interest (VOI)-based functional magnetic resonance imaging (fMRI) analysis. (**A**) Mean time-course of blood oxygenation level-dependent (BOLD) signals in the transition between rest and perceived flickering (PF) or rest and unperceived flickering (UF), averaged over the fMRI voxels corresponding to the subject-specific spectroscopic VOI. (**B**) BOLD percent change during the experimental conditions, averaged over the fMRI voxels corresponding to the subject-specific spectroscopic VOI. No statistically significant difference in BOLD response was found between the two conditions with conventional statistics, and Bayesan paired sample *t*-test indicated moderate evidence for the absence of difference between the conditions. (**C, D**) Average onset time as a function of peak intensity fraction and corresponding time-to-peak (i.e., at 100% peak intensity). There is a small, although not statistically significant trend for slower onset and longer time-to-peak of BOLD increase during the UF condition. (**E**) BOLD percent change averaged over the fMRI voxels corresponding to the Brodmann Area 17 (i.e., V1). Within V1, there is no difference between PF and UF condition (unpaired two-sample *t*-test, p = 0.72). (**F**) BOLD percent change averaged over the fMRI voxels corresponding to the Brodmann Areas 18 and 19 (e.g., including V2, V3a, V4v, and V5/MT). Within these areas, the response to PF is significantly larger than the corresponding response to UF (unpaired two-sample *t*-test, p = 0.008). *, statistically significant.

The online version of this article includes the following source data and figure supplement(s) for figure 1:

**Source data 1.** Demographics and functional magnetic resonance imaging (fMRI) study parameters.

**Source data 2.** Datasets and Matlab scripts for generating panels from *Figure 1* and associated figure supplements.

**Figure supplement 1.** Eye position and gaze displacement during functional magnetic resonance imaging (fMRI) sessions.

**Figure supplement 2.** Pupil size dynamics during functional magnetic resonance imaging (fMRI) sessions.

**Figure supplement 3.** Task performance during functional magnetic resonance imaging (fMRI) sessions.

**Figure supplement 4.** Correlation of task performance with eye position/pupil size during functional magnetic resonance imaging (fMRI) sessions.

**Figure supplement 5.** Perceived flickering (PF) versus unperceived flickering (UF) blood oxygenation level-dependent (BOLD) matching.

9% of BA19), was not different (0.44% ± 0.30% for PF vs 0.41% ± 0.25% for UF, paired sample *t*-test, p = 0.71) between the two conditions (*Figure 1B*). Bayesian analysis indicated moderate evidence for the absence of a difference between conditions (paired sample Bayesian *t*-test, BF01 = 3.2). The fMRI activations to PF and UF both peaked in V1 and distinctly spanned bilaterally in secondary visual areas ((*Figure 2A, B*), one-sample *t*-test, false discovery rate [FDR] corrected at cluster level, q < 0.05, voxel level p < 0.001).

## Different BOLD responses in secondary visual areas to PF and UF

To better characterize the effect of the two different stimulations, we estimated the main effect of the flickering frequency. The main effect of PF appeared in the lateral occipital cortices, but not in

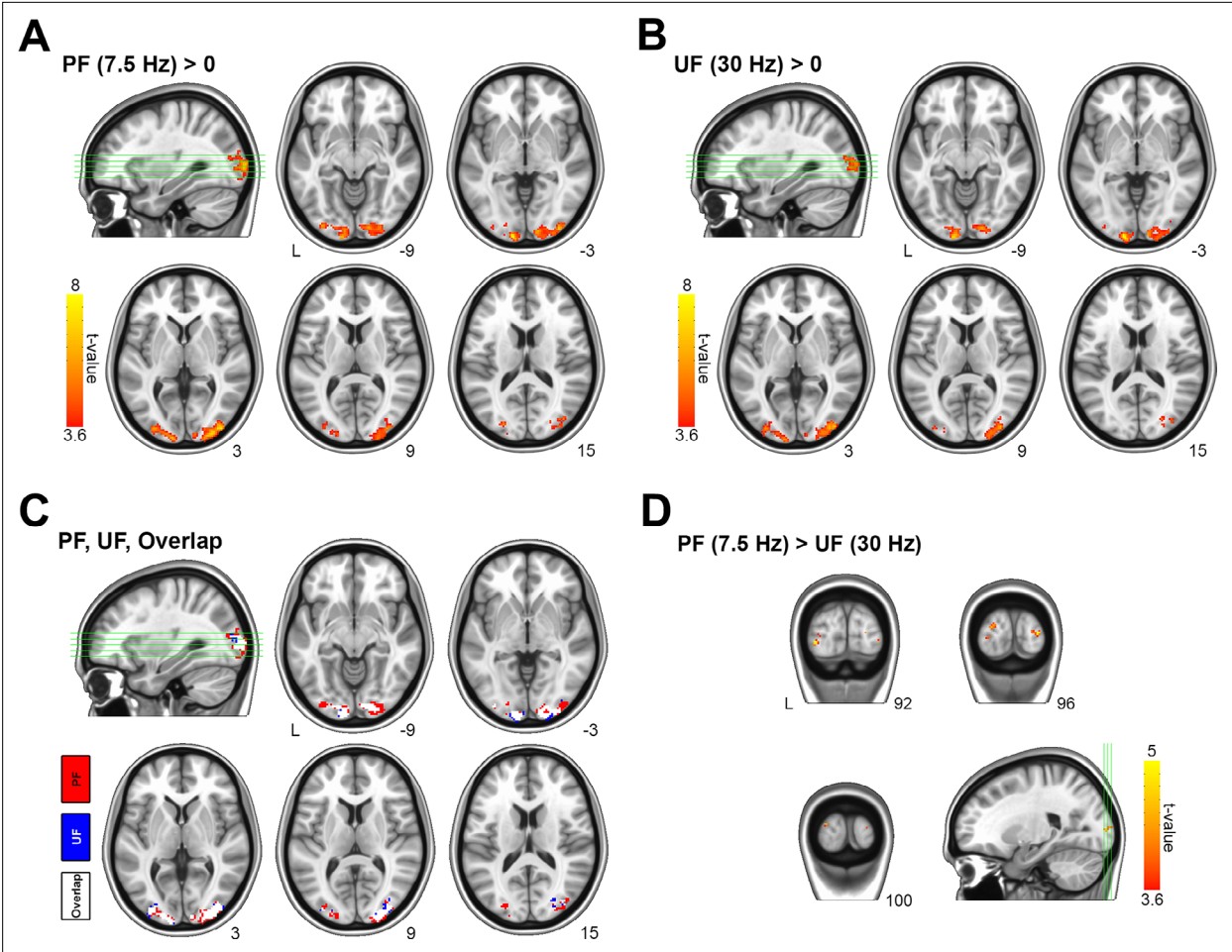

**Figure 2.** Main and differential effects of stimulation assessed by voxel-based functional magnetic resonance imaging (fMRI) analysis. (**A**) Statistical maps for group-averaged positive effect of the perceived flickering (PF) visual stimulation versus rest (i.e., PF > rest). (**B**) Statistical maps for group-averaged positive effect of the PF visual stimulation versus rest (i.e., unperceived flickering [UF] > rest). (**C**) Overlap between activation maps relative to PF and UF. (**D**) Differential effect of stimulation (PF> UF). The differential response related to perception is localized in the lateral occipital cortex (secondary visual areas), with no responding voxels inside V1. For comparison, the inverse differential effect of stimulation (UF > PF) has no significant responding voxels (not shown). All statistical activation maps are thresholded at p < 0.001, with a false discovery rate (FDR) correction at the cluster level (corresponding to $q_{FDR}$ <0.05), and overlaid on MNI template.

The online version of this article includes the following source data and figure supplement(s) for figure 2:

**Source data 1.** Cortical regions preferentially activated by perceived flickering (PF) compared to unperceived flickering (UF).

**Figure supplement 1.** Positions of regions and volumes of interest within the imaging field.

V1 (*Figure 2C and D*; *Figure 2—source data 1*). Bilaterally, 17.5% (i.e., 18 voxels out of 103) of the cortical volume preferentially activated by PF overlapped with the average spectroscopic VOI (*Figure 2—figure supplement 1*), corresponding to a contamination of approximately 0.5% (i.e., 18 voxels out of 3691). Although the electrophysiological activity in V1 could not be directly assessed in our experiment, based on the literature (*Logothetis et al., 2001*) we can obtain a rough surrogate of V1 output by evaluating fMRI signals in the secondary visual areas, which receive input directly from V1. While the average BOLD response in V1 (Brodmann Area 17) was similar (0.85% ± 0.45% vs 0.80 ± 0.42%, paired sample t-test, p = 0.72) for the two stimulation conditions (*Figure 1E*), the average BOLD change in secondary visual areas (Brodmann Areas 18 and 19) was significantly (0.61% ± 0.29% vs 0.39 ± 0.18%, paired sample t-test, p = 0.008) higher during PF than UF (*Figure 1F*), indicating a larger output from V1 during PF compared with UF. Thus, V1 exhibited the same BOLD signal despite known differences in visual processing for PF and UF (*Viswanathan and Freeman, 2007*).

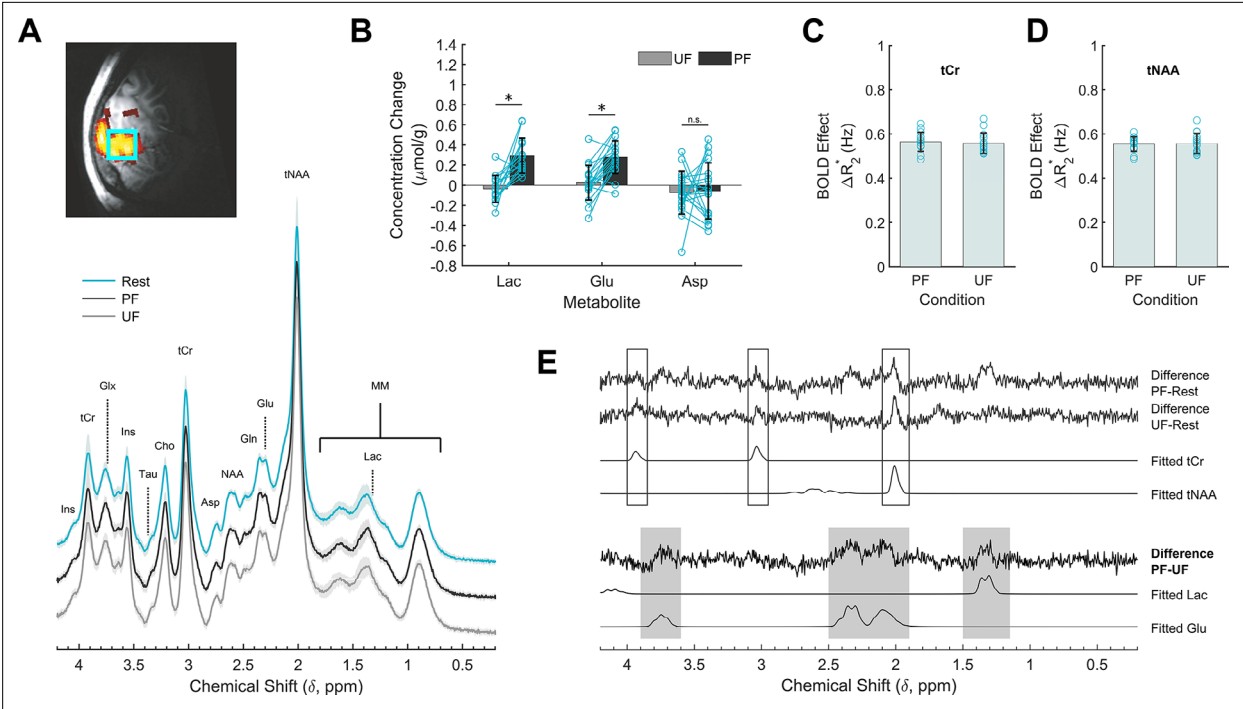

**Figure 3.** Effects of stimulation on the cortical metabolic profile assessed by 1H-fMRS (functional magnetic resonance spectroscopy) analysis. (**A**) Spectroscopic data acquired during resting (R, cyan) as well as perceived flickering (PF; black) and unperceived flickering (UF; gray) conditions, averaged across subjects. A single-subject representative voxel location is reproduced on a parasagittal view of the blood oxygen level-dependent (BOLD) activation and superimposed on the anatomical scan from the same subject. For visualization purposes, the processing of the spectra included frequency and phase correction of single transients, averaging, eddy currents correction, and Fourier transform. (**B**) Lactate, glutamate, and aspartate concentration changes during the stimulation conditions, relative to the rest conditions acquired immediately before. Data are averaged across subjects. There is significant increase in lactate (+28%) and glutamate (+3%) levels induced by PF stimulus, but not by UF stimulus. The concentration changes of the two metabolites were significantly different across the stimulation conditions, while there was no change for aspartate. (**C, D**) Spectral total creatine (tCr) and total *N*-acetylaspartate (tNAA) linewidth changes induced by the PF and UF stimuli show no statistically significant difference. (**E**) Differences between spectra acquired in the three experimental conditions. For reference, the corresponding LCModel fits are reported on the bottom for the Lac and Glu signals. tCr and tNAA singlets showed the expected BOLD-related features: there is a difference between stimulation and rest, but the difference spectra between the active conditions are within the noise. In the regions of lactate and glutamate the difference spectra between PF and rest and between PF and UF are similar, while they are clearly distinct from the difference spectra between UF and rest. *, statistically significant.

The online version of this article includes the following source data and figure supplement(s) for figure 3:

**Source data 1.** Demographics and 1H-fMRS (functional magnetic resonance spectroscopy) study parameters.

**Source data 2.** Modulations of metabolic profile of V1 during perceived flickering (PF) and unperceived flickering (UF) stimulations.

**Source data 3.** Datasets and Matlab scripts for generating panels from *Figure 3* and associated figure supplements.

**Figure supplement 1.** Eye position and gaze displacement during 1H-fMRS (functional magnetic resonance spectroscopy) sessions.

**Figure supplement 2.** Pupil size dynamics during 1H-fMRS (functional magnetic resonance spectroscopy) sessions.

**Figure supplement 3.** Task performance during 1H-fMRS (functional magnetic resonance spectroscopy) sessions.

**Figure supplement 4.** Correlation of task performance with eye position/pupil size during functional magnetic resonance imaging (fMRI) sessions.

**Figure supplement 5.** Quality of 1H-fMRS (functional magnetic resonance spectroscopy) spectra.

## Dissociation between metabolic and BOLD responses in V1 during loss of visual perception

To assess whether metabolism was sensitive to stimulus processing within V1 we performed single-voxel 1H-fMRS. The reproducible BOLD response in the occipital lobe elicited by the PF and UF stimulations allowed a very accurate VOI positioning for 1H-fMRS. High quality (water linewidth 7.2 ± 0.6 Hz) and artifact-free spectra were reliably obtained from almost all subjects (*Figure 3A*; *Figure 3—figure supplement 5*; *Figure 3—source data 1*). Compared with resting conditions, the lactate and glutamate concentrations within V1 increased by 0.29 ± 0.18 and 0.28 ± 0.16 µmol/g, respectively,

during the PF stimulation corresponding to an increase of about 28% and 3%, respectively, over the baseline (paired sample t-test, $q_{FDR}$ = 0.001), whereas they both remained at their basal levels (−0.04 ± 0.13 μmol/g, $q_{FDR}$ = 0.42 for lactate, and 0.03 ± 0.17 μmol/g, $q_{FDR}$ = 0.63 for glutamate) during the UF stimulation (*Figure 3B*). The lactate and glutamate responses were significantly different (paired sample t-test, $q_{FDR}$ = 0.01 for lactate and $q_{FDR}$ = 0.003 for glutamate) among the two stimulation conditions (*Figure 3*). No other metabolites among those quantified showed a reliable stimulation-dependent change (*Figure 3—source data 2*). We were unable to detect a reliable change for aspartate (paired sample t-test, $q_{FDR}$ = 0.98), a proposed index of MAS and oxidative metabolism (*Mangia et al., 2007a*).

To confirm our fMRI result of similar BOLD effect during PF and UF, we examined linewidth narrowing of total creatine (tCr) and total *N*-acetylaspartate (tNAA) signals. We found a stimulation-induced decrease of tCr and tNAA linewidth during both PF and UF (*Figure 3C, D*) that was not statistically different between the two conditions (0.56 ± 0.04 vs 0.56 ± 0.05 Hz for tCr, paired sample t-test, p = 0.70; 0.55 ± 0.03 vs 0.56 ± 0.05 Hz for tNAA, paired sample t-test, p = 0.89).

To substantiate our 1H-fMRS result, we determined the difference spectra between conditions, which mainly consisted of uncorrelated noise and only a few correlated residuals (*Figure 3E*). Specifically, the difference between PF and rest spectra showed a signal in the region corresponding to lactate, and in spectral regions corresponding to glutamate; both signals were absent in the difference between UF and rest spectra. The difference spectra also featured some narrow peaks corresponding to the main singlets of the spectrum, particularly tCr and tNAA, as a result of BOLD-induced line narrowing (*Zhu and Chen, 2001*). Similar residuals on tCr and tNAA were recognizable in the difference spectrum between UF and rest, but not in the difference spectrum between the two active conditions, again consistent with the evidence of a similar BOLD effect on spectral linewidth elicited by either of the stimulations. Overall, the only correlated signals that survived in the difference spectrum between PF and UF were lactate and glutamate, which strongly supports the significance of the concentration changes based on LCModel quantifications.

## Discussion

The cortical gray matter of the brain features one of the highest metabolic rates of all organ tissues of the human body. Although energy is recognized as a limiting factor for the human cerebral cortex (*Herculano-Houzel, 2011*; *Niven and Laughlin, 2008*), the increase in lactate concentration occurring upon sensory stimulation is not the result of limited oxygen availability (*Aalling et al., 2018*; *DiNuzzo, 2016*), as it is for skeletal muscle. Why the cerebral cortex upregulates glycolytic metabolism for sensory information processing is unknown, but it is well established that glycolysis serves specific neurophysiological and neurobiological purposes, such as axonal vesicle transport, vesicle recycling, action potential waveform modulation, reuptake of neuroactive compounds, and dendritic spine remodeling (reviewed in *DiNuzzo and Nedergaard, 2017*). Furthermore, lactate is known to be implicated in cognitive processes occurring during waking activity, like learning and memory (*Descalzi et al., 2019*; *Newman et al., 2011*; *Scavuzzo et al., 2020*; *Suzuki et al., 2011*), although the exact underlying mechanisms are still debated (*Dienel, 2019b*; *Steinman et al., 2016*).

In the present study, we asked whether alterations in visual perception are also reflected in metabolic changes within the primary visual cortex in humans. We report that stimulus perception affects the lactate and glutamate response in V1. The PF and UF stimulations elicited, by experimental design, the same average BOLD signals increase inside V1, indicating an equivalent degree of neurovascular coupling and possibly of local synaptic activity in the two experimental conditions (*Logothetis, 2008*). Yet, we observed a significant increase in the regional lactate level only during the PF stimulus, with no appreciable change of lactate during the UF stimulus compared with resting conditions. A temporal dissociation between BOLD and lactate changes has been previously reported during repeated photic stimulations (*Mangia et al., 2007b*). In that case, BOLD response was preserved, but not the lactate accumulation, possibly due to habituation of neuronal firing. Metabolic adaptation, in terms of glutamate levels, in the presence of constant BOLD and electrophysiological activity was also reported in epilepsy (*Peca et al., 2010*). These results suggest that the physiological mechanisms underlying BOLD signals and energy metabolism do not necessarily overlap under all experimental conditions.

Our results suggest that lactate and glutamate may be dissociated from BOLD changes when cortical input and output are differentially modulated by for example, intracortical (*Viswanathan and*

*Freeman, 2007*) or thalamocortical (*Min et al., 2020*) inhibition. In fact, an overall switch toward inhibition is expected to reduce the energy request of the brain, thus impacting on metabolic rates independently of BOLD response (*Mangia et al., 2009*). Although inhibition, in terms of GABA concentration, has been positively correlated with changes of the BOLD signal (*Chen et al., 2005*; *Donahue et al., 2014*; *Muthukumaraswamy et al., 2009*; *Northoff et al., 2007*), the dependence of GABAergic neurotransmission and corresponding BOLD changes upon temporal flickering frequency is difficult to predict, as the gating of sensory information is much likely supported by dishinibitory mechanisms (e.g., through vasoactive intestinal peptide-positive interneurons) (*Askew et al., 2019*; *Gasselin et al., 2021*; *Pi et al., 2013*; *Williams and Holtmaat, 2019*; *Yu et al., 2019*). Accordingly, changes in excitation/inhibition equilibrium have been proposed as a source of variability of the CBF/CMRO$_2$ coupling ratio (*Buxton et al., 2014*). In more general terms, the metabolic response seems capable of differentiating neural states that are intrinsically distinct, although they induce the same BOLD response (*Moradi et al., 2012*). Similar BOLD signal changes in correspondence to substantially different alterations in tissue lactate and glutamate level during visual flickering could not be predicted a priori unless assuming fundamental differences in information processing during stimulation. In particular, the firing rate of layer IV neurons that receive input from LGN is higher during UF than PF, as evidenced by the synchronization of these neurons to the stimulus frequency (*Gur and Snodderly, 1997*; *Herrmann, 2001*; *Norcia et al., 2015*; *Pastor et al., 2003*; *Regan, 1989*; *Vialatte et al., 2010*), yet we found that lactate and glutamate increases during PF, not UF. These findings indicate that the stimulation-induced effect on metabolite concentrations is happening downstream of the input stage in layer IV, and specifically during the intracortical processing involving output layers II/III and V. This conclusion is supported by the fact that high-frequency flickering (30–60 Hz) abolished MUA, a measure of neuronal spiking (putatively from layer II/III) (*Viswanathan and Freeman, 2007*), but not single neuron recordings from layer IV (*Gur and Snodderly, 1997*).

Using compartmentalized metabolic modeling, we have recently obtained evidence that the abovementioned outcome can be explained by distinct signaling mechanisms underlying spiking and synaptic activity (e.g., pre- and postsynaptic ionic currents) that are indeed frequency dependent (*DiNuzzo and Giove, 2012*; *DiNuzzo et al., 2014*). We previously reported that chromatic and achromatic flickering at the same temporal frequency elicited the same neurochemical response in V1 despite differences in the responding neuronal populations (i.e., blob vs interblob) (*Bednařík et al., 2018*). Together with the results that we report here, this observation suggests that lactate and glutamate, and hence aerobic glycolysis, are both sensitive to cortical processing (e.g., input/output or excitation/inhibition balance) rather than the targeted neuronal population. Our results can be interpreted in keeping with the concept that increased glutamate levels reflect an upregulation of the MAS (*Mangia et al., 2012*), although we were unable to report significant differences in aspartate levels likely due to insufficient sensitivity of MRS at 3T. Indeed, the decrease in aspartate during sensory stimulation is commonly observed at 7T, but to our knowledge has never been reported at 3T, most likely due to limited sensitivity of detection. In fact, the small absolute concentration of aspartate, its multiplet spectrum profile, and its overlap with other metabolites pose several challenges for detection of the small changes (around 0.1–0.2 mmol/l) occurring during a functional design. Similar considerations actually apply to glucose, which also was not reported in this study to decrease as observed in 7T studies (*Mangia et al., 2007a*). The NADH produced by the metabolism of glucose to lactate has to be transported from the cytosol to the mitochondria to be used for energy production within the electron transport chain. Among other components of the MAS, such process requires that glutamate is transported from cytosol to mitochondria and concomitantly aspartate is counter-transported from mitochondria to cytosol through the aspartate–glutamate carrier (AGC1/Aralar). The AGC1/Aralar carrier-mediated transport is a rate-limiting step within the MAS and therefore it lags behind the concomitant increase in tricarboxylic acids cycle and NAD$^+$/NADH-redox shuttling in mitochondria. In agreement with existing literature (*Hertz and Chen, 2017a*; *Hertz and Rothman, 2017b*; *Lin et al., 2012*; *Mangia et al., 2012*; *Mangia et al., 2007a*), we interpret the increase in glutamate (and decrease in aspartate) concentration as reflecting an increase in the rate of MAS, with accumulation of glutamate at the cytosolic side and reduction of aspartate at the mitochondrial side. Notably, the MAS does not necessarily correlate with mitochondrial respiration and cerebral blood flow, as evidenced by the findings that (1) oxidative metabolism and CMRO$_2$ are enhanced at both low and high flickering frequencies (*Viswanathan and Freeman, 2007*), and (2) glutamate and aspartate,

but not lactate, correlate with BOLD signals during low frequency (~8 Hz) visual stimulation (*Bednařík et al., 2018*; *Ip et al., 2017*). To what extent aerobic glycolysis and MAS can dissociate from oxidative phosphorylation in the brain remains to be established, but previously published data strongly indicate that the increase in lactate and glutamate levels is the consequence of the surge in glycolysis and the ensuing increase of cytosolic NADH:NAD$^+$ ratio taking place during neuronal activation (*Díaz-García et al., 2017*).

While the similar BOLD response in V1 is a direct consequence of our experimental design, we additionally found that secondary visual areas were recruited to a larger extent during PF stimulus than UF stimulus, supporting the argument that their activation is necessary for perception (*Lamme et al., 2000*; *Salminen-Vaparanta et al., 2019*; *Tong, 2003*). The preferential response of secondary visual areas to perceived stimuli that we observe here broadly confirms previous results of a strong BOLD activity in ventral ('visual-for-perception' processing) and dorsal ('visual-for-action' processing) streams during visible stimuli and a reduction of their activity in conditions of invisible stimulation (*Hesselmann and Malach, 2011*). However, since our stimulation protocol is associated by design with higher activation of output layers of V1 and hence higher input to V2/V3, we cannot conclusively and unambiguously establish that the higher activation of secondary visual areas is causally linked to perception. In fact, the very notion that BOLD activity in secondary visual areas actually reflects V1 output is uncertain because we have no direct measurement to firmly support such assumption, although it is very tightly linked to the unique architecture of the visual system.

The role of the primary visual cortex in stimulus perception is an intense area of research (*Gail et al., 2004*; *Leopold and Logothetis, 1996*; *Maier et al., 2008*; *Polonsky et al., 2000*; *Wunderlich et al., 2005*). Previous fMRI studies investigating the dependence of V1 BOLD activity on visual perception produced controversial results, which might be related to the difficulty of disentangling perception from spatial attention. Indeed, it has been shown that attention, but not perception, modulates the BOLD signal in human V1 (*Watanabe et al., 2011*). In fact, perception was found to enhance the BOLD response within V1 for specific experimental designs (e.g., flash suppression of perception) (*Yuval-Greenberg and Heeger, 2013*). Our study employed a substantially different stimulation paradigm that specifically allowed us, by adjusting the stimulation contrast, to induce the same BOLD response, and inductively an equivalent mean degree of synaptic activity, within V1 under the two experimental conditions (*Logothetis, 2008*). Most importantly, we did not use any additional stimulus (e.g., visual masking) to suppress or modulate visual perception. On the contrary, we used a simple way to modulate perception for extended periods of time (required by 1H-fMRS) while maintaining attention at a nearly constant level, as evidenced by pupillometry and task performance. In particular, we employed the well-known dependence of BOLD response in V1 to flickering, which at full-contrast peaks at 4–8 Hz and settles around 70% of its maximal value even at frequencies above 30 Hz (*Thomas and Menon, 1998*), that is, in the absence of flickering perception. Previous literature reported that the peak activations in V1 and secondary visual areas are stimulation dependent and occur at distinct temporal frequencies (4 or 8 Hz in V1, and 20 or 40 Hz in secondary visual areas for chromatic or luminance flickering, respectively) (*Chai et al., 2019*; *Fox et al., 1986*; *Kastner et al., 2004*; *Kwong et al., 1992*; *Singh et al., 2000*). Interestingly, the preferred frequency of secondary visual areas is just below the CFF for chromatic flickering (~25 Hz) and luminance flickering (~50 Hz) (*Shady et al., 2004*). In our experiments, the isoluminant chromatic flickering stimulation at 30 Hz is above the CFF and accordingly we found that the BOLD response in secondary visual areas drops substantially compared with the concurrent BOLD response in V1. Overall, by using visual stimulations below and above the CFF and adjusting the contrast of the low-frequency stimulus we were able to modulate perception alongside BOLD activity in secondary visual areas with a unchanged BOLD activity in V1. It should be noted that subtle differences in the qualitative nature of the BOLD signal response might exist between the two conditions, although the trends toward longer onset time and time-to-peak during UF compared with PF were not statistically significant.

Our study has some limitations. For instance, the fMRI measurements have been obtained using 30 s epochs, while the fMRS measurements have been obtained using 4 min epochs. Long acquisition times are required for achieving a good sensitivity of the 1H-fMRS measurements and a reliable metabolite quantification, even at magnetic fields of 3T. Nonetheless, results in both humans (*Mangia et al., 2007b*) and rats (*Just et al., 2013*) have shown that prolonged (i.e., minutes) stimulations elicit a persistent BOLD response with minimal attenuation. Although we cannot exclude a certain

degree of neuronal adaptation, we confirmed that the BOLD effect during the fMRS acquisition was indeed relatively stable as determined by the linewidth changes of tCr and tNAA signals. Another limitation to consider is represented by the large volume used for 1H-fMRS measurements. It could be possible that the observed changes in lactate levels include distinct neuronal populations showing nonuniform responses. Indeed, although the main differential effect of frequency was located far from the calcarine sulcus, it also involved some mid hemispheric areas presumably included in the spectroscopic voxel. As an illustration, heterogeneity in the response to stimulation within V1 might be due to, for example, eccentricity (*Chai et al., 2019*), which we however minimized by using a 3° foveal stimulation. There is also an hemispheric lateralization for the spatial frequencies (*Kauffmann et al., 2014*), but this is not likely to apply to our study, as we used the same radial checkerboard (i.e., including many spatial frequencies) for both PF and UF. The finding that the changes in BOLD signal during PF and UF were nearly double in the anatomical (i.e., Brodmann atlas-based) V1 compared with the spectroscopic VOI indicates a substantial partial-volume effect. The associated loss of sensitivity entails that the lactate concentration change that we report here might in fact be underestimated. Furthermore, we cannot exclude that feedback input to V1 from secondary visual areas might be taking place during stimulus perception (*Watkins et al., 2006*). However, under our experimental design we were able to fully control BOLD response in V1 by only acting on stimulus contrast, without altering the subjective perception of the stimuli, which indicates that the BOLD activity in V1 largely reflected stimulus features. Finally, we based the positioning of the spectroscopic VOI solely on the online estimate of the stimulus-induced BOLD signal change. In particular, we could not perform online retinotopic mapping for driving the positioning of the spectroscopic VOI, as this would have increased the acquisition protocol duration to an impractical length for guaranteeing good attention levels and absence of motion during the 1H-fMRS session. In any case, partial-volume effects are unavoidable because the STEAM sequence we used requires a box-shaped VOI, which would have not allowed us to entirely avoid the inclusion of tissue outside V1, regardless of the availability of the retinotopic map.

## Conclusion

In this study, we report for the first time that visual perception has a measurable metabolic effect on V1. Although we could not establish any causal relation between metabolism and perception, our findings imply that BOLD-fMRI and 1H-fMRS are complementary techniques. Specifically, they are capable of highlighting different aspects of neural activation and stimulus processing, for example, under conditions in which synaptic and spiking activity are partially disentangled due to an alteration of excitation/inhibition patterns and, in turn, local input/output balance.

Although we focused on the visual pathways, it is conceivable that our results can be translated to other sensory modalities. For example, tactile (*Laureys et al., 2002*) or auditory (*Boly et al., 2004*; *Laureys et al., 2000*) stimulations during vegetative state can still activate primary somatosensory cortex (S1) or primary auditory cortex (A1), respectively, in the absence of perception and without the involvement of higher-order associative areas.

We suggest that the cortical metabolic profile might be an indicator of sensory perception, in keeping with the dynamics of lactate and glutamate across the sleep–wake cycle (*Aalling et al., 2018*; *DiNuzzo and Nedergaard, 2017*; *Naylor et al., 2012*; *Naylor et al., 2011*) as well as with the relation between lactate and arousal (*Zuend et al., 2020*). Elevated brain lactate and glutamate levels are associated with wakefulness and memory formation, which naturally require the processing of incoming sensory stimuli, like the control exerted by the central visual pathways for either gating or filtering out behaviorally relevant or irrelevant visual information. As such, the metabolic responses to perceived, but not unperceived, sensory stimulation could be enabling factors for learning and memory, as indicated by the relevance of aerobic glycolysis and lactate in synaptic plasticity mechanisms (*Bueschke et al., 2021*; *Descalzi et al., 2019*; *DiNuzzo, 2016*; *Harris et al., 2019*; *Herrera-López et al., 2020*; *Jourdain et al., 2018*; *Kobayashi et al., 2019*; *Lundquist et al., 2021*; *Margineanu et al., 2018*; *Scavuzzo et al., 2020*; *Wang et al., 2019*; *Yang et al., 2014*). In particular, aerobic glycolysis and lactate might reflect cortical information processing and, in turn, intracortical communication, in agreement with the relation between regional metabolic rates of glucose utilization and resting-state network dynamics in the cerebral cortex (*Jamadar et al., 2020*; *Noack et al., 2017*; *Spetsieris et al., 2015*; *Su et al., 2018*; *Thompson, 2018*).

## Materials and methods

### Setup

MR measurements were performed on a 3T head-only scanner (Siemens Allegra, Erlangen, Germany), equipped with high performance gradients (amplitude 40 mT/m, rise time 100 µs). A custom-built quadrature surface coil (loop and butterfly design, Rapid Biomedical, Rimpar, Germany) was used for both RF transmission and detection. The coil design traded usable volume (*Figure 2—figure supplement 1*) for peak sensitivity. For imaging, first- and second-order shim terms were adjusted using the automatized Siemens routine based on field map acquisition and fitting. For MRS, shimming was optimized using FASTMAP with EPI readout (*Gruetter and Tkác, 2000*) and manually refined when necessary to reach a water linewidth of less than 9 Hz.

### Subjects

A total of 32 healthy volunteers were initially enrolled for this study after they gave informed consent, according to the Helsinki declaration and to European Union regulations, and following the approval by the Ethics Committee of the Fondazione Santa Lucia IRCCS (Rome). Exclusion criteria included any kind of previous neurological or psychiatric disease and impaired visual acuity. Thirteen subjects were discarded either upon online fMRI processing (see below) or in early quality control on data, because of motion (3) or because the most activated area was in unfavorable position with respect to the surface-coil sensitive volume (10). Nineteen healthy volunteers (10 females, 9 males; age 25 ± 4 years), mean ± standard deviation (SD; age range 20–35 years) were thus considered for this study. Sample size calculations performed before the study assumed a two-tail paired *t*-test design, a power of 0.95 and were based on an effect magnitude for lactate change (photic stimulation vs rest) of 0.20 ± 0.15 µmol/g as estimated in our previous works at 7T (*Bednařík et al., 2018*). Reduced sensitivity of 3T vs 7T was empirically accounted for by larger VOI and extended data averaging (144 transients per condition, 10 ml VOI at 3T vs 64 transients per condition, 8 ml VOI at 7T). The resulting required sample size of 10 was roughly doubled to account for multiple comparisons.

### Visual stimulation

Visual stimulation consisted in a radial checkerboard flickering either at 7.5 Hz (PF stimulation) or at 30 Hz (UF stimulation). The alternating frames included either a gray-green or a gray-red checkerboard. The stimuli were programmed in Cogent 2000 version 1.29 working under Matlab 2006b (The Mathworks, Natick, MA, USA) and delivered using an MR-compatible fMRI system with stereo 3D goggles (VisuaStim Digital, Resonance Technology, Inc, Northridge, CA, USA). Subjects were fitted with the VisuaStim video goggles (resolution: SXGA 1280 × 1024 pixels, refresh rate: 60 Hz, field of View: 30° horizontal × 24° vertical, White Luminance: 70 cd/m² max, contrast ratio: intrinsic 100:1 measured per VESA FPDM standard).

### Experimental protocol

Visual stimulations were presented in eight (fMRI) or four (fMRS) epochs, with stimulation epochs (either PF or UF) interleaved by rest (isoluminant, uniform gray images) epochs. Since the CFF is higher for luminance than for chromatic flickering (*Jiang et al., 2007*), before each session the brightness of green squares during the UF condition (i.e., isoluminant condition) was adjusted interactively by the subject, who piloted increasing and decreasing brightness ramps and was instructed to identify the brightness level corresponding to loss of luminance flickering perception of the resulting yellow. The green level was then set midway between the two perceptual vanishing levels. The stimulus contrast was adjusted in preliminary acquisitions on five subjects, in order to induce a comparable BOLD response to PF and UF in V1 (*Figure 3A*) During the fMRI sessions epochs lasted 30 s each (total 4 min), while for fMRS acquisitions epochs were 4 min long (total 16 min). Each subject underwent 1 fMRI and 2 fMRS runs (10 min apart, without moving the subject from inside the scanner); the order of PF and UF conditions was counterbalanced within each subject, and the initial stimulation type was randomized between subjects.

## Task

In order to maintain visual fixation and keep a constant attentional state, the subjects were asked to focus on a central target (a cross) and to press a button whenever the target rotated. Subjects were specifically instructed to maintain their attention on the fixation cross rather than focusing on reaction times (i.e., speed to push the button). The number of rotations was constant across the different epochs (3 for each fMRI epoch and 24 for each 1H-fMRS epoch, or approx. 1 rotation every 10 s), while the exact timing of the rotation was pseudorandomized (range 2–18 s).

## Anatomical and fMRI data acquisition

Each study started with an anatomical acquisition (MPRAGE T1-weighted volumetric scan, resolution $1.2 \times 1.2 \times 1.2$ mm$^3$, para-axial slices, in-plane field of view (FOV) $190 \times 70$ mm$^2$, TE = 4.38 ms, TR = 2000 ms, TI = 910 ms, FA = 8°). Then, one fMRI session (pseudorandomized order of stimulation across subjects) was acquired (gradient echo with EPI readout, resolution $2.2 \times 2.2 \times 2.2$ mm$^3$, 26 para-axial contiguous slices, FOV $190 \times 70$ mm$^2$, TE = 30 ms, TR = 1500 ms, FA = 70°). fMRI scans were processed online for subsequent MRS voxel positioning (online processing included motion correction, smoothing, cross-correlation with a square-wave model; the two scans following each condition change were discarded in order to reduce the effects of BOLD signal transients. Online processing was discarded after voxel positioning).

To confirm the absence of any detectable brain pathology in our subjects, T1- and T2-weighted anatomical scans were acquired with a standard volume birdcage coil after the end of the functional scans. Anatomical scans included an MPRAGE acquisition (resolution $1.0 \times 1.0 \times 1.0$ mm$^3$, para-axial slices, in-plane FOV $256 \times 160$ mm$^2$, TE = 2.48 ms, TR = 2150 ms, TI = 1000 ms, FA = 8°), that was later used during the postprocessing for normalization purposes.

## fMRS data acquisition

The spectroscopic voxel (size $25 \times 20 \times 20$ mm$^3$) was localized in the most activated area within V1, based on both anatomical scan and results of the online fMRI processing. The voxel was located either left or right of the interhemispheric fissure to minimize the cerebrospinal fluid fraction in the VOI. Two MRS sessions were acquired with an optimized, in-house written STEAM sequence (TE = 7 ms, TM = 50 ms, TR = 3000 ms, FA = 70°) which included outer volume saturation and VAPOR water suppression (*Tkác et al., 2001*; *Tkác et al., 1999*). An eight-step phase cycle was used; transients were averaged within each phase cycle, and each phase cycle was saved separately for further processing. Water unsuppressed data were acquired from the same voxel for eddy currents compensation (*Klose, 1990*). In order to minimize T1 weighting, the flip angle was kept below the calculated Ernst angle in both fMRI and fMRS acquisitions.

## Pupillometry

In order to monitor attentional state with a physiological parameter, we acquired pupillometry data using an eye-tracking system (Applied Science Laboratories, model 504) equipped with a remote pan/tilt optic infrared module and a video camera that was custom adapted for use in the scanner. Subject gaze position and pupil size data were processed as previously described (*DiNuzzo et al., 2019*).

## fMRI data processing

fMRI (offline) processing was performed with routines from SPM12 (Wellcome Trust Centre for Neuroimaging, UCL) working under Matlab 2018b, AFNI (*Cox, 1996*), and FSL5 (*Jenkinson et al., 2012*), and with custom Matlab routines. fMRI data were realigned to their mean image to compensate for head movements. Realigned images were then normalized to the MNI template ($2 \times 2 \times 2$ mm$^3$) by using the nonlinear transformation calculated on the MPRAGE acquired with the volume coil, after a linear coregistration that used the surface-coil MPRAGE image as intermediate step to best match the volume-coil MPRAGE to the fMRI series. Normalized images were spatially smoothed with an isotropic 4 mm full width at half-maximum Gaussian kernel. A general linear model (GLM) analysis with boxcars functions convolved with the hemodynamic response function was applied to localize the regions responding to the PF and UF conditions. The GLM model included a high-pass filter (128 Hz) and the six rotational and translational parameters obtained from the realignment step.

For the VOI-based analysis, images underwent the same processing pipeline with the only exception that the analysis was performed in the subject-specific space, thus, no spatial normalization step was applied.

Head motion during fMRI acquisitions was evaluated using the framewise displacement, which was calculated as the L1-norm of the realignment-derived parameters after converting angles to linear displacements (*Power et al., 2012*).

## 1H-fMRS data processing

MRS data were preprocessed using jMRUI 5.2 (*Naressi et al., 2001*) and custom Matlab routines. Data were corrected for residual eddy currents, individually phased and frequency shifted to compensate for $B_0$ drifts, and averaged in blocks corresponding to each rest or stimulation epoch. The first eight transients of each epoch, that is the first full phase cycle (24 s) were discarded to avoid metabolic transients (*Mangia et al., 2007a*). Subsequent phase cycles were inspected individually. They consistently showed good water suppression and no trace of lipidic contamination. A few 8-transient spectra (maximum one in each epoch) featured anomalous line broadening, line splitting or otherwise reduced quality, putatively related to subject motion or deep inspiration, and were discarded before averaging. Each epoch spectrum was thus the average of 64–72 transients. The resulting averages were finally quantified using LCModel 6.3-1 (*Provencher, 1993*) with a tailored basis set. Basis metabolites included alanine, aspartate (Asp), creatine (Cr), γ-aminobutyric acid (GABA), glutamine (Gln), glutamate (Glu), glycine, glycerylphosphorylcholine, glutathione (GSH), lactate (Lac), *myo*-inositol (Ins), *N*-acetylaspartate (NAA), *N*-acetylaspartylglutamate (NAAG), phosphocholine, phosphocreatine, phosphorylethanolamine, *scyllo*-inositol, and taurine (Tau). Glucose, an important marker of energy metabolism, whose changes have also been reported in previous 7T studies (*Bednařík et al., 2015*; *Mangia et al., 2007a*), was not included in the basis set due to highly unreliable quantification observed in preliminary tests. Metabolite spectra were simulated using GAVA (*Soher et al., 2007*), including information on the sequence pulse program. The basis set included also a macromolecular signal, that was acquired on each subject in the occipital region, using a double inversion recovery approach (STEAM, TI1 = 1700 ms, TI2 = 520 ms, TE = 7 ms, TM = 50 ms, TR = 2000 ms, FA = 90°) (*de Graaf et al., 2006*), that resulted in almost complete metabolite nulling, averaged between subjects, and then modeled with Hankel–Lanczos singular value decomposition. LCmodel quantifications with Cramér–Rao lower bounds above 30% were discarded, except for Lac for which the threshold was set at 35%. Since this study is focused on epoch-to-epoch metabolic changes, absolute quantification with water referencing was not performed to avoid the uncertainty associated with motion between scans, relaxation, and partial-volume corrections for white matter and gray matter in the VOI. Metabolites were instead normalized to the tCr signal amplitude fitted over each run, which was here assumed to correspond to 7.5 μmol/g (see *de Graaf, 2007*). Eleven metabolites were quantified in at least 15 subjects (80% of participants). These included aspartate, tCr, GABA, glutamate, glutamine, glutathione, lactate, *myo*-inositol, tNAA (or NAA plus NAAG), total choline, and taurine. In order to take BOLD-induced spectral linewidth alterations into account during the fMRS acquisitions, we determined the BOLD effect as the kernel size (in Hz) that minimized the difference spectra (windowed around the reference metabolite peak, or 2.82–3.14 ppm for tCr and 1.75–2.25 ppm for tNAA) between the stimulated epoch (either PF or UF) and the preceding resting epoch, under the assumption that the levels (i.e., area under the peak) of tCr and tNAA remain constant across epochs (*de Graaf, 2007*). All spectra were then averaged according to three categories: rest, PF, and UF conditions, and differences were calculated after performing linewidth matching between stimulated and corresponding rest conditions (*Mangia et al., 2007a*).

## Statistics

For pupillometry and task-performance results, statistical comparisons were made using Student's *t*-test and one-way ANOVA on the rest, UF, and PF conditions. No post hoc test was necessary. Correlations between mean pupil diameter or gaze displacement and task performance were computed as Kendall's tau coefficients and corrected for multiple comparisons using FDR.

For fMRI results, correction for multiple comparisons in functional voxel-based analysis was performed using FDR correction. Resulting clusters were also checked through Monte Carlo simulation using the AFNI tool Alphasim (*Cox, 1996*) after estimation of residuals smoothness. Comparison

between conditions was performed in Matlab 2018b with ANOVA, unpaired, or paired $t$-test as appropriate. Significance of difference of BOLD response in the spectroscopy voxel (*Figure 1B*) was additionally assessed with Bayesian paired $t$-test with level of evidence set to 3.

For fMRS results, statistical analysis was restricted to those reliably quantified metabolites associated with energy metabolism that showed consistent functional changes in previous fMRS studies (*Bednařík et al., 2015*; *Bednařík et al., 2018*; *Lin et al., 2012*), namely Lac, Glu, and Asp. Metabolite concentration changes referred to the corresponding resting epoch and between different active conditions were tested using paired sample $t$-tests, with FDR correction for nine multiple comparisons.

Data were presented as the mean ± SD. A p value, or a $q_{FDR}$ value where relevant, of less than 0.05 was considered as statistically significant.

## Data and materials availability

All data that support the findings of this study are available from the corresponding author upon signing a MTA that would include a list of authorized researchers and the commitment to no further distribute the materials. Data are not available in a public repository at the time of this publication because of constraints originally set by the Ethics Committee and included in the informed consent signed by participants.

The study was developed using SPM12 (https://www.fil.ion.ucl.ac.uk/spm/software/spm12/), LCmodel (http://s-provencher.com/lcmodel.shtml), jMRUI (http://www.jmrui.eu/), and custom scripts. Custom MATLAB code for fMRI and 1H-fMRS data processing is available in figure supplements (*Figure 1—source data 2* and *Figure 3—source data 3*) and in online public repository (https://github.com/dmascali/mni2atlas) *DiNuzzo, 2021* copy archived at swh:1:rev:d05afff7081ddade66afc39c28d76c8cda1d6d39.

## Acknowledgements

The authors wish to thank Edward J Auerbach for the FASTMAP implementation on Siemens platform, provided by the University of Minnesota under a C2P agreement, and for his help with the setup of offline shim currents calculation. Siemens Healthineers is acknowledged for providing source code and information on the shim coils.

## Additional information

### Funding

| Funder | Grant reference number | Author |
|---|---|---|
| Ministero della Salute | Ricerca Corrente | Federico Giove |
| Max Planck Institute for Biological Cybernetics | Open Access funding | Gisela E Hagberg |

The funders had no role in study design, data collection, and interpretation, or the decision to submit the work for publication.

### Author contributions

Mauro DiNuzzo, Conceptualization, Data curation, Formal analysis, Investigation, Software, Validation, Visualization, Writing - original draft, Writing - review and editing; Silvia Mangia, Conceptualization, Methodology, Supervision, Writing - original draft, Writing - review and editing; Marta Moraschi, Data curation, Formal analysis, Investigation, Methodology, Resources, Validation, Writing - review and editing; Daniele Mascali, Data curation, Formal analysis, Investigation, Methodology, Resources, Software, Visualization, Writing - review and editing; Gisela E Hagberg, Conceptualization, Funding acquisition, Methodology, Project administration, Writing - review and editing; Federico Giove, Conceptualization, Data curation, Formal analysis, Funding acquisition, Investigation, Methodology, Project administration, Software, Supervision, Writing - review and editing

### Author ORCIDs

Mauro DiNuzzo http://orcid.org/0000-0003-0181-5597

Silvia Mangia http://orcid.org/0000-0001-6341-4516
Marta Moraschi http://orcid.org/0000-0003-4789-6636
Daniele Mascali http://orcid.org/0000-0003-1269-6060
Gisela E Hagberg http://orcid.org/0000-0003-2176-7086
Federico Giove http://orcid.org/0000-0002-6934-3146

## Ethics

This study included human subjects and was performed by the authors in compliance with all applicable ethical standards, including the Helsinki declaration and its amendments, institutional/national standards, and international/national/institutional guidelines. The study was approved by the Ethics Committee of Fondazione Santa Lucia (Rome). All subjects gave informed consent before being enrolled in the study.

## Decision letter and Author response

Decision letter https://doi.org/10.7554/eLife.71016.sa1
Author response https://doi.org/10.7554/eLife.71016.sa2

## Additional files

### Supplementary files

• Transparent reporting form

### Data availability

The study was developed using SPM12 (https://www.fil.ion.ucl.ac.uk/spm/software/spm12/), LCmodel (http://s-provencher.com/lcmodel.shtml), jMRUI (http://www.jmrui.eu/) and AFNI (https://afni.nimh.nih.gov/). Data used for all the figures and for Tables 2-3 is available as source data to each element. Source data include also custom Matlab code for processing related to each figure. The raw data include sensitive data. The raw dataset cannot be made available in a public repository because of constraints originally set by the Ethics Committee and included in the informed consent signed by participants. Raw data that support the findings of this study are available from the corresponding author upon signing a MTA that would include: a list of authorized researchers; a commitment to not disclose the raw data to persons not included in the list; and a commitment to destroy the raw data when legitimate use is finished. Commercial use of the raw data is not permitted.

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
