## [Editor Report]

The authors demonstrate an intriguing dissociation of neurovascular (as measured with BOLD-fMRI) and neurometabolic (measured with fMRS) responses during perception. This is a thought-provoking study that makes one wonder about the neurophysiological origin of the signals we measure with human neuroimaging, especially fMRI. It will therefore be of great interest to the broad community of neuroimagers, as well as perception researchers.

---

## [Decision Letter]

**Decision letter after peer review:**

Thank you for submitting your article "Perception affects the brain's metabolic response to sensory stimulation" for consideration by *eLife*. Your article has been reviewed by 3 peer reviewers, including Peter Kok as the Reviewing Editor and Reviewer #1, and the evaluation has been overseen by Tirin Moore as the Senior Editor. The following individuals involved in review of your submission have agreed to reveal their identity: David Norris (Reviewer #2); Polytimi Frangou (Reviewer #3).

The reviewers have discussed their reviews with one another, and the Reviewing Editor has drafted this to help you prepare a revised submission. The reviewers agreed that the study is of interest but had serious concerns about the interpretation of some of the main results.

Essential revisions:

1. The possibility exists that you have a significant partial volume contamination of V2 in your spectroscopy voxel, and that this activates more in the PF than the UF condition. In fact, you yourselves raise a potential partial voluming issue in the fMRS measurement that seems important to consider, given the differential BOLD signal in nearby regions (V2 and V3). This effect may not fully explain your results because you have such large increases in Glu and Lac, but it may exaggerate them. In this context, you used the Freesurfer coordinates to locate the primary visual areas. Is there any reason why you did not perform retinotopic mapping? There can be a large variation in the size of early visual regions between subjects (https://www.nature.com/articles/ncomms3201). Retinotopic mapping would have made it possible to exclude V2 from your spectroscopic voxel on an individual basis. Is there something you can do to address this conclusively?

2. Although you matched the level of BOLD response, the qualitative nature of the BOLD activation within V1 will probably differ. In the Discussion you point out that the firing rate of layer IV neurons will be higher during UF than PF. This would imply a higher level of activity in the input layer during UF than PF. As the total BOLD activity was matched then there will by necessity have been higher activity in the layers outside layer IV for PF, which by definition implies a higher activation in the output layers of V1 and hence a higher input to V2. The significant increase in activation in V2 cannot be unambiguously ascribed to perception as your design forces a higher level of activity in the output layers of V1. Please discuss how this affects the interpretability of your findings.

3. The acquisition of MRS data necessitates long acquisition times and block designs. Early work on lactate emphasised that it was only detectable for prolonged stimulation. Work from your lab (Mangia et al. 2007) suggests that lactate levels are highest during the first period of stimulus than later. Do you believe that the lactate signal results from prolonged stimulation of a (sub)population of neurons or is it intrinsic to the process? Please discuss the potential role of lactate in more detail, for instance by including a short historical overview regarding the interpretation of the lactate signal, going back to the Prichard paper. Also some discussion of the role of lactate as a substrate for neurons would be interesting.

4. Lin et al. (2012) found a reduction in aspartate in a similar experiment but performed at 7T. In your experiment Asp remains unchanged. Is either result in any way consistent with an upregulation in MAS? Differences in glutamate could be suggested to reflect higher levels of excitatory synaptic activity: did you consider this?

5. In Figure 2B, it looks like BOLD dynamics may differ between the slow and fast flicker blocks, even if the mean amplitude did not. So perhaps there are some more subtle BOLD differences between conditions that are not sufficiently explored. (E.g. an increased BOLD response to perceived flicker early on, follow by adaptation; Figure 2B looks potentially in line with this.) Is it possible to look into this further? It is unclear whether the time-courses between PF and UF were compared statistically. During visual stimulation under wakefulness vs. anaesthesia, onset time and time to peak of the BOLD response differ between the two states (e.g. Dinh, T. N. A. et al. 2021, Neuroimage). The authors suggest that the metabolic processes described here are in line with lactate and glutamate dynamics across the sleep-wake cycle; therefore, it would be interesting to explore the differences in the time-course of the BOLD response for perceived vs. unperceived stimulation in primary and secondary visual areas.

Please see the recommendations made by the individual reviewers for further suggestions and more detail on some of these points.

*Reviewer #1 (Recommendations for the authors):*

In Figure 2B, it looks like BOLD dynamics may differ between the slow and fast flicker blocks, even if the mean amplitude did not. So perhaps there are some more subtle BOLD differences between conditions that the authors do not sufficiently explore. (E.g. an increased BOLD response to perceived flicker early on, follow by adaptation; Figure 2B looks potentially in line with this.) Is it possible to look into this further?

Also, the mean BOLD response in the MRS voxel has quite large error bars (Figure 2C); is the absence of a difference between conditions reliable, or due to low SNR/large intersubject variability? Some Bayesian statistics could help.

Regarding the potential partial voluming issue raised above, is there something the authors can do to address this conclusively?

In general, establishing a null effect is of course challenging. For instance, can we be sure a different fMRI sequence would also have yielded a null effect?

*Reviewer #2 (Recommendations for the authors):*

You have shown that in early visual cortex the metabolic response profile as measured by proton spectroscopy differs between two experimental conditions, even when the total BOLD response is matched. Given our knowledge of the origins of the BOLD signal change, and of the metabolite signal changes that you record this is a highly plausible result. I do have some questions about aspects of your experimental design and interpretation.

1. Although you matched the level of BOLD response, the qualitative nature of the BOLD activation within V1 will probably differ. In the Discussion you point out that the firing rate of layer IV neurons will be higher during UF than PF. This would imply a higher level of activity in the input layer during UF than PF. As the total BOLD activity was matched then there will by necessity have been higher activity in the layers outside layer IV for PF, which by definition implies a higher activation in the output layers of V1 and hence a higher input to V2. The significant increase in activation in V2 cannot be unambiguously ascribed to perception as your design forces a higher level of activity in the output layers of V1.

2. It would have been interesting to have measured CBF as well as BOLD. By matching BOLD alone, the possibility remains that in one condition (probably PF) both CBF and CMRO2 could be elevated relative to the other condition (UF) leading to equal BOLD responses but different underlying metabolic rates.

3. The acquisition of MRS data necessitates long acquisition times and block designs. Early work on lactate emphasised that it was only detectable for prolonged stimulation. Work from your lab (Mangia et al. 2007) suggests that lactate levels are highest during the first period of stimulus than later. My somewhat philosophical question, is do you believe that the lactate signal results from prolonged stimulation of a (sub)population of neurons or is it intrinsic to the process? In this context I would also have appreciated a short historical overview regarding the interpretation of the lactate signal, going back to the Prichard paper. Also some discussion of the role of lactate as a substrate for neurons would be interesting.

4. Lin et al. (2012) found a reduction in aspartate in a similar experiment but performed at 7T. In your experiment Asp remains unchanged. Is either result in any way consistent with an upregulation in MAS? Differences in glutamate could be suggested to reflect higher levels of excitatory synaptic activity: did you consider this?

5. You used the Freesurfer coordinates to locate the primary visual areas. Is there any reason why you did not perform retinotopic mapping? There can be a large variation in the size of early visual regions between subjects (https://www.nature.com/articles/ncomms3201). Retinotopic mapping would have made it possible to exclude V2 from your spectroscopic voxel on an individual basis. The possibility exists that you have a significant partial volume contamination of V2 in your spectroscopy voxel, and that this activates more in the PF than the UF condition. I do not believe that this effect explains your results because you have such large increases in Glu and Lac but it may exaggerate them.

6. Why wasn't the experiment done at 7T? You would have had a smaller MRS voxel and a fuller metabolic profile?

7. If you would like to do a better experiment (though I am not insisting that this should be done) then instead of trying to match the activation level in V1, I would perform the experiment at different contrast levels. I would also perform retinotopic mapping to ensure that partial voluming from V2 etc. is negligible. You could test then if the intercepts of the Lac/glu concentration as a function of stimulus intensity are linear with the same intercept, implying both are involved in metabolic activity, or have different intercepts implying that they are involved in different processes.

*Reviewer #3 (Recommendations for the authors):*

1) The authors briefly touch upon the time-course of the BOLD response in Figure 2B. It is unclear whether the time-courses between PF and UF were compared statistically. During visual stimulation under wakefulness vs. anaesthesia, onset time and time to peak of the BOLD response differ between the two states (e.g. Dinh, T. N. A. et al. 2021, Neuroimage). The authors suggest that the metabolic processes described here are in line with lactate and glutamate dynamics across the sleep-wake cycle; therefore, it would be interesting to explore the differences in the time-course of the BOLD response for perceived vs. unperceived stimulation in primary and secondary visual areas.

2) Could the authors elaborate on how upregulation of the MAS would result in increased MRS-visible glutamate? My understanding is that upregulation of the MAS would result in increased glutamate transportation into the mitochondrial matrix – how would this impact glutamate in the extracellular space that is presumably what MRS picks up?

3) Removing the statistics from the plots would further declutter the figures.

4) The order of the figures should be reconsidered. While the authors provide an impressive array of convincing controls for attention, these are not the focus of the paper and should not occupy Figure 1. These analyses would be best suited as figure supplements to Figure 4, showing that the differences observed cannot be attributed to attentional modulation. Similarly, Table 1 should be split into two tables as figure supplements to Figures 2 and 4, respectively or provided as source data. Table 2 could be provided as source data.

The first question the authors answer is whether there are differences in the BOLD signal between the two frequency conditions. This is answered in (1) Figure 3B – no differences in fMRI BOLD within V1 for PF vs UF and (2) 2C/3C/3D – differences within BA18/19 but not BA17 or the MRS voxel for PF vs UF.

The authors provide further supporting evidence to secondary questions about (i) areas activated for PF vs rest and UF vs rest (2B, 2C) and (ii) whether they overlap (3A). These secondary questions (as well as Figure 2A that refers to methodology) can be added as figure supplements without distracting the reader from the key findings.

5) You show in figure S4 the different ROIs (BA17, BA18, BA19) and your MRS voxel. It would be very informative if you also overlayed the MRS voxel on the PF>UF activation of Figure 3B and quantified the overlap.

[Editors' note: further revisions were suggested prior to acceptance, as described below.]

Thank you for resubmitting your work entitled "Perception is associated with the brain's metabolic response to sensory stimulation" for further consideration by *eLife*. Your revised article has been reviewed by 2 peer reviewers and the evaluation has been overseen by Tirin Moore as the Senior Editor, and a Reviewing Editor.

The manuscript has been improved but there are some remaining issues that need to be addressed before we can accept your paper for publication, as outlined below:

1) In the manuscript, it is still unclear how you performed normalisation to the concentration of creatine, as we do not understand the use of "assumed concentration of creatine" in the VOI (line 612). We agree that creatine would be the appropriate reference, rather than water, (lines 610-611 could explain what the "inherent uncertainty" is) and would expect that measured concentration of metabolites is referenced to the measured concentration of creatine. Indeed, this is what we understand from your response letter, but it's not clear from your manuscript text. To be clear, the concern is mainly that your methods are easily understood and replicable, rather than a concern about their validity.

The following points are suggestions for clarifications.

2) Thank you for rearranging the figures and adding figure supplements. We appreciate your concerns about the two main figures and the increased number of supplements for Figure 1. A potential solution could be splitting the fMRI results in two figures:

Figure 1: VOI-based fMRI results; that is the current Figure 1 and the figure supplements that include control analyses (Figure 1 – supplements 1,2,3,4,5)

Figure 2: voxel-based fMRI results; that is a composite figure of Figure 1 – supplement 6 and Figure 1 – supplement 7, with Figure 1 – supplement 8 as supplement

Figure 3: fMRS results; as is, that is the current Figure 2 with its supplements.

3) Thank you for providing a detailed explanation to our question about the interpretation of glutamate increase. Lines 336-349 in your revised text contain important interpretational information and would benefit from being rewritten in a more accessible way for the wider readership your paper will attract.

4) Our understanding from lines 615-620 is that you iteratively apply linewidth-matching until the amplitude of the difference spectra is minimised; that is because you assume that the amplitude in the difference spectra would arise from either BOLD effects or true metabolite differences. This process is not very clear in the text and it would be helpful to clarify the steps.

---

## [Author Response]

Essential revisions:1. The possibility exists that you have a significant partial volume contamination of V2 in your spectroscopy voxel, and that this activates more in the PF than the UF condition. In fact, you yourselves raise a potential partial voluming issue in the fMRS measurement that seems important to consider, given the differential BOLD signal in nearby regions (V2 and V3). This effect may not fully explain your results because you have such large increases in Glu and Lac, but it may exaggerate them. In this context, you used the Freesurfer coordinates to locate the primary visual areas. Is there any reason why you did not perform retinotopic mapping? There can be a large variation in the size of early visual regions between subjects (https://www.nature.com/articles/ncomms3201). Retinotopic mapping would have made it possible to exclude V2 from your spectroscopic voxel on an individual basis. Is there something you can do to address this conclusively?

We agree on these points. As noticed, we ourselves have raised the possibility for partial-volume effect. Accordingly, the reported changes in lactate and glutamate during the PF condition could be somewhat overestimated, assuming that the sign of the changes in V1 and V2/V3 are congruent between each other. We also agree that subject level retinotopic mapping would have improved experimental design. However, we could not perform online retinotopic mapping for driving the spectroscopic voxel positioning within our 1H-fMRS protocol because of time constraints. Indeed, acquiring and processing retinotopic information, and transferring back the maps to the scanner, would have lengthened the acquisition protocol duration to more than 90 min, an impracticable length for guaranteeing good attention levels and absence of motion during the fMRS session. Furthermore, partial volume would have been in any case unavoidable considering that the STEAM sequence we used requires a rectangular box-shaped voxel (not an arbitrary shape, although there have been proposals in such direction, like ASSESS sequence or 2D-SSE/SPEN pulses). In particular, a box-shaped voxel would have not allowed to entirely avoid inclusion of tissue outside V1, regardless of the availability of the retinotopic map. This is due to anatomical considerations, i.e. the three-dimensional shape of the visual cortex. Therefore, we based the positioning of the spectroscopic voxel solely on the online (i.e., in-scanner) determination of the stimulation-induced BOLD-fMRI signal change. In this revision, we have stressed the lack of retinotopic mapping as a limitation of the study.

2. Although you matched the level of BOLD response, the qualitative nature of the BOLD activation within V1 will probably differ. In the Discussion you point out that the firing rate of layer IV neurons will be higher during UF than PF. This would imply a higher level of activity in the input layer during UF than PF. As the total BOLD activity was matched then there will by necessity have been higher activity in the layers outside layer IV for PF, which by definition implies a higher activation in the output layers of V1 and hence a higher input to V2. The significant increase in activation in V2 cannot be unambiguously ascribed to perception as your design forces a higher level of activity in the output layers of V1. Please discuss how this affects the interpretability of your findings.

We agree that the qualitative nature of the BOLD signal within V1 might be different during the PF and UF conditions, which also implies different ratios between the neural activity in input vs. output layers. We also agree that the changes we have reported in BOLD signal within V2/V3 (and by the way, also the changes in metabolite levels within V1) cannot be unambiguously ascribed to perception. In fact, we tried to avoid any discussion about “causality”, but we rather focused on “association”. We realized that in some circumstances we might have used ambiguous terminology, thus in this revision we have paid particular attention not to over-interpret our findings, also emphasizing the inappropriateness of implying causality between our results and perception. Finally, we slightly changed the title accordingly.

3. The acquisition of MRS data necessitates long acquisition times and block designs. Early work on lactate emphasised that it was only detectable for prolonged stimulation. Work from your lab (Mangia et al. 2007) suggests that lactate levels are highest during the first period of stimulus than later. Do you believe that the lactate signal results from prolonged stimulation of a (sub)population of neurons or is it intrinsic to the process? Please discuss the potential role of lactate in more detail, for instance by including a short historical overview regarding the interpretation of the lactate signal, going back to the Prichard paper. Also some discussion of the role of lactate as a substrate for neurons would be interesting.

Previous 1H-fMRS experiments from our group and other labs clearly showed that there is a net lactate increase in the stimulated cortical tissue in the order of approx. 20% that develops within about 1 minute and then settles there to a new steady-state. Partial volume here could partly explain the slow dynamics of the lactate signal, as electrode-based lactate measurements in the stimulated cat visual cortex have reported that extracellular lactate concentration rises on the sub-second time scale in a manner which is robustly time-locked with the visual stimulus (Li and Freeman, J Neurochem 2015). Unfortunately, based on current literature it is not possible to assign the lactate increase to a specific neuronal population. In fact, there is no way of assigning the lactate increase to any cellular population, including astroglia. The latter argument prevents discussing the interpretation of lactate as a substrate for neurons, because any underlying astrocyte-to-neuron or neuron-to-astrocyte lactate shuttle remains unproven in vivo (see, for example, Dienel, J Neurosci 2017). Moreover, lactate has been proposed to have a role as a signaling molecule rather than (or in addition to) a metabolic substrate (DiNuzzo, Front Integr Neurosci 2016). We have even advanced new hypothesis on the production and flux of lactate in the brain during stimulation related to glycogen (see, for example, DiNuzzo et al., JCBFM 2010, 2019). Overall, while the stimulation-induced increase in brain lactate is a well-established concept, the exact mechanisms responsible for its production as well as its fate (whether metabolic or other) is unknown. Therefore, we would avoid reiterating the examination about the role of lactate in the brain, which we feel would distract from the main focus of the present paper, also considering that the discussion is already quite long and dense. However, as noted in XXX at page XXX, we invite the interested reader to refer to the papers published by our and other groups on the lactate debate.

4. Lin et al. (2012) found a reduction in aspartate in a similar experiment but performed at 7T. In your experiment Asp remains unchanged. Is either result in any way consistent with an upregulation in MAS? Differences in glutamate could be suggested to reflect higher levels of excitatory synaptic activity: did you consider this?

The involvement of activation-induced changes of glutamate and aspartate observed in previous works has been indeed ascribed to MAS (Mangia et al., JCBFM 2007; Mangia et al., Neurochem Res 2012; Lin et al., JCBFM 2012; Hertz and Chen, Front Integr Neurosci 2017; Hertz and Rothman, Biology 2017). Yet, a decrease in aspartate during sensory stimulation is commonly observed at 7T, while to our knowledge has never been reported at 3T (including the current study), most likely due to lower detection sensitivity. Therfore we would like to refrain from interpreting the lack of an observed Asp change in our study as evidence of a physiological process.

The suggested idea of a direct relation between glutamate and higher levels of excitatory synaptic activity is interesting. However, the amount of released glutamate (on average, 1 umol/g in 1 minute) is comparably minor compared with the intracellular pool (10 umol/g in neurons), and part of the released glutamate has been shown to undergo oxidation in astrocytes (i.e., disappear), with the oxidized fraction increasing non-linearly with the extracellular glutamate levels, and hence neurotransmission (McKenna, Front Endocrinol 2013). Furthermore, NMR visibility of glutamate might change during neurotransmission due to different mobility of the glutamate molecules when they are released compared to when they are densely packed into synaptic vesicle, which would additionally complicate the interpretation of a direct relation between glutamate and neurotransmission per se. In light of such observations, discussing the possibility that different findings among studies may be ascribed to different levels of excitatory activity could be unnecessarily speculative, although this is certainly a fascinating idea. In this revision we mention the main reason for why aspartate is not commonly reported at 3T, and we also clarify our interpretation that glutamate and aspartate changes relates to MAS.

5. In Figure 2B, it looks like BOLD dynamics may differ between the slow and fast flicker blocks, even if the mean amplitude did not. So perhaps there are some more subtle BOLD differences between conditions that are not sufficiently explored. (E.g. an increased BOLD response to perceived flicker early on, follow by adaptation; Figure 2B looks potentially in line with this.) Is it possible to look into this further? It is unclear whether the time-courses between PF and UF were compared statistically. During visual stimulation under wakefulness vs. anaesthesia, onset time and time to peak of the BOLD response differ between the two states (e.g. Dinh, T. N. A. et al. 2021, Neuroimage). The authors suggest that the metabolic processes described here are in line with lactate and glutamate dynamics across the sleep-wake cycle; therefore, it would be interesting to explore the differences in the time-course of the BOLD response for perceived vs. unperceived stimulation in primary and secondary visual areas.

We agree that the average BOLD signals that we report during PF and UF might contain subtle differences, possibly related to processing of visual information in V1. In this revision we statistically compare the single points of the average BOLD timecourses during PF vs. UF. We also evaluate the onset time and time-to-peak (adding a supplementary figure). None of the statistical tests was significant, although there was a trend for slightly delayed onset during UF compared with PF.

Unfortunately, due to the lack of retinotopic mapping we could not extract the BOLD time-course separately in primary vs. secondary visual areas (please note that this kind of analysis is carried out in the subject-space) unless we performed multiple back-and-forth normalization steps that would have produced unreliable results at the single-subject level.

Please see the recommendations made by the individual reviewers for further suggestions and more detail on some of these points.

We integrate the information provided above within the responses given below to the individual referees’ comments.

Reviewer #1 (Recommendations for the authors):In Figure 2B, it looks like BOLD dynamics may differ between the slow and fast flicker blocks, even if the mean amplitude did not. So perhaps there are some more subtle BOLD differences between conditions that the authors do not sufficiently explore. (E.g. an increased BOLD response to perceived flicker early on, follow by adaptation; Figure 2B looks potentially in line with this.) Is it possible to look into this further?

The argument raised by the referee is interesting, and we thank the referee for letting us look deeper in the BOLD dynamics. We performed one-way ANOVA on the average BOLD time-courses, but we found no statistically different time points. We believe that more data would be necessary to identify such subtle differences, if any. We however mention in the text that subtle differences in BOLD dynamics might still exist between conditions.

Also, the mean BOLD response in the MRS voxel has quite large error bars (Figure 2C); is the absence of a difference between conditions reliable, or due to low SNR/large inter-subject variability? Some Bayesian statistics could help.

We acknowledge that there is inter-subject variability, partially mitigated by the pairing. Following the suggestion of the reviewer, we added a Bayesian paired t-test between the conditions to help assess the results reliability. Bayesian analysis indicates that there is moderate evidence for the absence of difference between conditions

(BF01>3). We accordingly updated the text. Unfortunately, we can’t address this issue in a conclusive manner, which is now added as a limitation of the study. Please see our response in the “Essential revisions” above.

In general, establishing a null effect is of course challenging. For instance, can we be sure a different fMRI sequence would also have yielded a null effect?

We agree with the reviewer that a null effect is intrinsically difficult to prove. Without prospectively acquiring relevant data, it is hard to exclude the possibility that other fMRI sequences may have found differences among conditions. However, we still consider this scenario unlikely, as generally speaking other fMRI sequences (such as spin-echo EPI or ASL sequences) have lower sensitivity than the EPI sequence used here, albeit they may have higher specificity to the yet-unknown mechanisms that may generate a difference in response.

Reviewer #2 (Recommendations for the authors):You have shown that in early visual cortex the metabolic response profile as measured by proton spectroscopy differs between two experimental conditions, even when the total BOLD response is matched. Given our knowledge of the origins of the BOLD signal change, and of the metabolite signal changes that you record this is a highly plausible result. I do have some questions about aspects of your experimental design and interpretation.1. Although you matched the level of BOLD response, the qualitative nature of the BOLD activation within V1 will probably differ. In the Discussion you point out that the firing rate of layer IV neurons will be higher during UF than PF. This would imply a higher level of activity in the input layer during UF than PF. As the total BOLD activity was matched then there will by necessity have been higher activity in the layers outside layer IV for PF, which by definition implies a higher activation in the output layers of V1 and hence a higher input to V2. The significant increase in activation in V2 cannot be unambiguously ascribed to perception as your design forces a higher level of activity in the output layers of V1.

The referee raises a very important point, that we incorporated in our revised manuscript. We now stressed that our findings are “associated” with stimulus perception, without necessarily a causal relationship. Nonetheless, a higher activity of output layers is a necessary correlate of perception. We understand that everything here would be speculative, so we do not elaborate further on this point in the manuscript.

2. It would have been interesting to have measured CBF as well as BOLD. By matching BOLD alone, the possibility remains that in one condition (probably PF) both CBF and CMRO2 could be elevated relative to the other condition (UF) leading to equal BOLD responses but different underlying metabolic rates.

We agree with the reviewer that measuring CBF in addition to BOLD may provide valuable insights. The suggestion of the referee is interesting and can be done in future investigation.

3. The acquisition of MRS data necessitates long acquisition times and block designs. Early work on lactate emphasised that it was only detectable for prolonged stimulation. Work from your lab (Mangia et al. 2007) suggests that lactate levels are highest during the first period of stimulus than later. My somewhat philosophical question, is do you believe that the lactate signal results from prolonged stimulation of a (sub)population of neurons or is it intrinsic to the process? In this context I would also have appreciated a short historical overview regarding the interpretation of the lactate signal, going back to the Prichard paper. Also some discussion of the role of lactate as a substrate for neurons would be interesting.

Please see our response in the “Essential revisions” above.

4. Lin et al. (2012) found a reduction in aspartate in a similar experiment but performed at 7T. In your experiment Asp remains unchanged. Is either result in any way consistent with an upregulation in MAS? Differences in glutamate could be suggested to reflect higher levels of excitatory synaptic activity: did you consider this?

Please see our response in the “Essential revisions” above.

5. You used the Freesurfer coordinates to locate the primary visual areas. Is there any reason why you did not perform retinotopic mapping? There can be a large variation in the size of early visual regions between subjects (https://www.nature.com/articles/ncomms3201). Retinotopic mapping would have made it possible to exclude V2 from your spectroscopic voxel on an individual basis. The possibility exists that you have a significant partial volume contamination of V2 in your spectroscopy voxel, and that this activates more in the PF than the UF condition. I do not believe that this effect explains your results because you have such large increases in Glu and Lac but it may exaggerate them.

Please see our response in the “Essential revisions” above.

6. Why wasn't the experiment done at 7T? You would have had a smaller MRS voxel and a fuller metabolic profile?

Because we were interested in confirming whether the typical changes observed at 7T are robustly observed on a more widespread clinical platform (3T), while implementing at the same time an original study design for addressing putative dissociation between BOLD-fMRI and fMRS. The point raised by the referee is also related to our MRS findings about aspartate. Please see our response in the “Essential revisions” above.

7. If you would like to do a better experiment (though I am not insisting that this should be done) then instead of trying to match the activation level in V1, I would perform the experiment at different contrast levels. I would also perform retinotopic mapping to ensure that partial voluming from V2 etc. is negligible. You could test then if the intercepts of the Lac/glu concentration as a function of stimulus intensity are linear with the same intercept, implying both are involved in metabolic activity, or have different intercepts implying that they are involved in different processes.

Such an experiment, as suggested by the referee, would be of exceptional value, but it would also be extremely costly and time-consuming. In fact, repeating the same measurements at different contrast levels would require for subjects to undergo excessively prolonged fMRI and MRS sessions (it took around 80 minutes to acquire the individual data sets used in the present study), especially at 3T where 4 min of averaging was needed for reliable metabolite quantification of each spectrum. A somewhat feasible version of such study may be designed for future investigations to be preferably conducted at higher fields.

Reviewer #3 (Recommendations for the authors):1) The authors briefly touch upon the time-course of the BOLD response in Figure 2B. It is unclear whether the time-courses between PF and UF were compared statistically. During visual stimulation under wakefulness vs. anaesthesia, onset time and time to peak of the BOLD response differ between the two states (e.g. Dinh, T. N. A. et al. 2021, Neuroimage). The authors suggest that the metabolic processes described here are in line with lactate and glutamate dynamics across the sleep-wake cycle; therefore, it would be interesting to explore the differences in the time-course of the BOLD response for perceived vs. unperceived stimulation in primary and secondary visual areas.

We thank the referee for this interesting comment. Following the present and another referee’s observation, we have now compared the BOLD time-courses during PF and UF. We did not find any statistically significant difference, nor onset-time and time-to-peak were significantly different. However, the suggestion that onset-time and time-to-peak might be able to distinguish between the two conditions is intriguing and we will look into it in future experiments with more statistical power.

2) Could the authors elaborate on how upregulation of the MAS would result in increased MRS-visible glutamate? My understanding is that upregulation of the MAS would result in increased glutamate transportation into the mitochondrial matrix – how would this impact glutamate in the extracellular space that is presumably what MRS picks up?

The referee is right. The up-regulation of the MAS results in an increased rate of transport of glutamate in mitochondria via the aspartate-glutamate carrier (AGC1), which also counter-transport aspartate out of the mitochondria. However, such transport requires concomitant exchange between malate and α-ketoglutarate via the oxoglutarate carrier (OGC), together with two dehydrogenation and two transamination reactions. When the overall MAS activity increases in order to process more glycolysis-derived NADH, all the various transporters and enzymes also accelerate. However, AGC1 is a rate-limiting step in MAS (i.e., it is the slower component), which results in a decrease in intramitochondrial aspartate and an increase in cytosolic glutamate (as explained in Mangia et al., Neurochem Res 2012; see also references therein). Notice that trans-mitochondrial malate transport is always possible (even independently of alphaketoglutarate counter-transport) due to the presence of dicarboxylate carrier (DIC, mediating malate exit from mitochondria) and citrate carrier (CIC, mediating malate entry into mitochondria with counter-transport of citrate).

3) Removing the statistics from the plots would further declutter the figures.

We have now removed statistics from the figure plots, as suggested by the referee.

4) The order of the figures should be reconsidered. While the authors provide an impressive array of convincing controls for attention, these are not the focus of the paper and should not occupy Figure 1. These analyses would be best suited as figure supplements to Figure 4, showing that the differences observed cannot be attributed to attentional modulation. Similarly, Table 1 should be split into two tables as figure supplements to Figures 2 and 4, respectively or provided as source data. Table 2 could be provided as source data.The first question the authors answer is whether there are differences in the BOLD signal between the two frequency conditions. This is answered in (1) Figure 3B – no differences in fMRI BOLD within V1 for PF vs UF and (2) 2C/3C/3D – differences within BA18/19 but not BA17 or the MRS voxel for PF vs UF.The authors provide further supporting evidence to secondary questions about (i) areas activated for PF vs rest and UF vs rest (2B, 2C) and (ii) whether they overlap (3A). These secondary questions (as well as Figure 2A that refers to methodology) can be added as figure supplements without distracting the reader from the key findings.

We thank the referee for these useful comments. In the present revision, we arranged the material to better focus on the main results of the study. We generally agree with the referee that some tables and figures are better suited for being supplementary figure or source data, and we actually followed all referee’s suggestions, yet we would prefer to maintain our logical flow in introducing our results. As a note, by moving most of the material as figure/data supplement as suggested by the referee, we ended up in having only two main figures in the manuscript.

5) You show in figure S4 the different ROIs (BA17, BA18, BA19) and your MRS voxel. It would be very informative if you also overlayed the MRS voxel on the PF>UF activation of Figure 3B and quantified the overlap.

This has been done. With the chosen statistics, the contamination of the PF>UF voxels represent 0.5% of the spectroscopic VOI. We also included a new figure supplement, as suggested by the referee.

[Editors' note: further revisions were suggested prior to acceptance, as described below.]

The manuscript has been improved but there are some remaining issues that need to be addressed before we can accept your paper for publication, as outlined below:1) In the manuscript, it is still unclear how you performed normalisation to the concentration of creatine, as we do not understand the use of "assumed concentration of creatine" in the VOI (line 612). We agree that creatine would be the appropriate reference, rather than water, (lines 610-611 could explain what the "inherent uncertainty" is) and would expect that measured concentration of metabolites is referenced to the measured concentration of creatine. Indeed, this is what we understand from your response letter, but it's not clear from your manuscript text. To be clear, the concern is mainly that your methods are easily understood and replicable, rather than a concern about their validity.

We agree that the description was a bit unclear. As highlighted by the Editors, the concentration of total creatine has been assumed constant. This is a standard approach in the brain MRS literature, albeit for absolute quantification water referencing (ie, assuming water content constant) is currently preferred. Thus, we have normalized the spectra to the tCr signal amplitude, fitted over the whole run. Results were converted into concentrations using tCr values taken from the literature. We changed the text in the relevant section, keeping also into account your remark about imprecise wording (“inherent uncertainty”).

The following points are suggestions for clarifications.2) Thank you for rearranging the figures and adding figure supplements. We appreciate your concerns about the two main figures and the increased number of supplements for Figure 1. A potential solution could be splitting the fMRI results in two figures:Figure 1: VOI-based fMRI results; that is the current Figure 1 and the figure supplements that include control analyses (Figure 1 – supplements 1,2,3,4,5)Figure 2: voxel-based fMRI results; that is a composite figure of Figure 1 – supplement 6 and Figure 1 – supplement 7, with Figure 1 – supplement 8 as supplementFigure 3: fMRS results; as is, that is the current Figure 2 with its supplements.

We have separated the VOI-based and the voxel-based fMRI results into two figures, as indicated. In addition to the suggestions of the Editors, we have also moved a Data source that was previously in Figure 1, because it turned out to be relevant to the new Figure 2.

3) Thank you for providing a detailed explanation to our question about the interpretation of glutamate increase. Lines 336-349 in your revised text contain important interpretational information and would benefit from being rewritten in a more accessible way for the wider readership your paper will attract.

We have slightly rewritten the portion of the text indicated by the Editors to make it more accessible. Although we have attempted to simplify the text, a detailed explanation would require an in-depth description of the malate-aspartate shuttle, which we feel is beyond the scope of the interpretational point that we want to make about glutamate and aspartate, considering that we have already discussed these arguments in other publications (e.g., Mangia et al., 2012).

4) Our understanding from lines 615-620 is that you iteratively apply linewidth-matching until the amplitude of the difference spectra is minimised; that is because you assume that the amplitude in the difference spectra would arise from either BOLD effects or true metabolite differences. This process is not very clear in the text and it would be helpful to clarify the steps.

The Editors are right, we assume that the amplitude in the difference spectra arises from either BOLD effects or true metabolite differences. Yet, we did not minimize the difference spectra corresponding to the entire spectral width, but only the spectral window containing the metabolite chosen to estimate the BOLD effect, whether total creatine (tCr) or total N-acetyl-aspartate (tNAA). Indeed, it is commonly assumed that the concentration of these metabolites does not change across conditions (rest versus stimulation). This assumption entails that the corresponding area under the curve remains constant across conditions, whereas the line width possibly changes (i.e., due to the BOLD effect). Under these conditions, our iterative approach guarantees that the adjustments of spectral line width that minimize the difference between spectra at rest and during stimulation coincides with the BOLD effect. We have modified the text to clarify our approach.